## [Peer Review File · Communications Chemistry]

Reviewers' comments:

Reviewer #1 (Remarks to the Author):

In this manuscript, the authors reported the identification of a set of iron cluster anions from experiment and the corresponding DFT characterization. Experimentally, they observed the distinct activity of the iron clusters against the uptake of molecular oxygen. Although results are very interesting, some points need to be discussed prior to publication.

It is well-known that iron has four naturally-occurring stable isotopes and oxygen has three naturally-occurring stable isotopes, and the corresponding results have never been discussed in the manuscript, especially for the MS spectra, and this is really needed. The combination of these isotopes might explain the noises around the main signal, and more detailed and systematic analysis are needed.

To explain the observed stability, they carried out DFT computations and found very surprising magnetic properties. For the most stable and abundant Fe₁₇O₁₀(-) cluster, they calculated the reaction of Fe₁₇O₈(-) + O₂ = Fe₁₇O₁₀(-) and a huge reaction energy of about 8 eV, which is as high as that of Fe₁₇(-) + O₂ = Fe₁₇O₂(-). If Fe₁₇O₁₀(-) is really so stable as suggested, the authors must calculate the further reaction of Fe₁₇O₁₀(-) + O₂ = Fe₁₇O₁₂(-) to check if this reaction is endergonic or exergonic. If it is endergonic, it is fine with the proposal, however, if it is exergonic, another explanation is needed.

As the reaction takes place at given temperature, they should use the thermal corrected Gibbs free energy instead of the computed relative energy to discuss the stability. Probably, temperature plays the decisive role in the stability of the cluster.

Other points

(1) The working temperature of the experiment was not given, and according to the DFT results in Figure 4f, the Fe₁₃(-) reacts with O₂ has a small barrier of 0.54 eV, and this is indeed not very high, and one can expect that at certain temperature, this barrier will vanish from the oxidized cluster directly. The reaction energy is as large as that of Fe₁₇(-) about 7-8 eV.

(2) For DFT, the BPW91 might be B3PW91, please check this carefully. In addition, the authors might check the basis set dependence for the energy and magnetisms by using 6-311+G* basis set (also adding polarization, and this is necessary for both Fe and O atoms, and single-point calculations should be acceptable, at least for the most stable structures).

(4) At the very beginning of results, they pointed out that they used different doses of oxygen reactants. However, this is not true, since they used always 10% in He.

In summary, major revision is needed.

Reviewer #2 (Remarks to the Author):

The paper reports for the first time a very interesting magic iron oxide cluster Fe₁₇O₁₀- among

other studied iron oxide anions. Experimental observations are supported by the results of computations using several reliable methods.

I recommend publication after the paper is revised according to the following remarks.

1. Page 2. Line 2 “Typically, FeO is antiferromagnetic” - This is true for the singlet state, but in the ground 5-Delta state, the total spin magnetic moments of both Fe and O atoms coupled parallelly according to the results of DFT-GGA calculations (Theor Chem Acc (2003) 109:298–308).

Line 32. “iron atoms are separated by oxygen atoms but without direct Fe-Fe bonds”. It is not the case for (FeO)_n clusters in cited Ref. [30] where the Fe-Fe bonds present. The cluster (FeO)₁₄ is ferromagnetic, which is difficult to explain.

2. Page 3. Line 63. Is it possible that the smaller Feⁿ⁻ and FeO_mⁿ⁻ are readily coalesce, and this is the reason for the shift into large masses in the mass spectra?

3. Figure 2. The total spin magnetic moment of the Fe₁₇⁻ is 53 μB. First, the Fe₁₇O₁₀⁻ anion has the odd number of electrons, and this moment should be odd. Second, no increase in the total spin magnetic moment due to oxidation was yet observed to the best of my knowledge. Probably, the increase is because the Fe_n host cluster is in an excited state stabilized by oxygens.

4. Page 7. Line 115. (from -4 to 56 μB), The total spin magnetic moment of a cluster equals to the difference between the number of electrons in the spin majority and spin minority representations, expressed in μB. Thus, no negative value is possible.

5. Figure 4. Only the first step is considered, while the final clusters are Fe₁₃O₈⁻ and Fe₁₇O₁₀⁻. There are many steps ahead, and it is hard to judge on them having only the first step done.

Reviewer #3 (Remarks to the Author):

The manuscript entitled “Fe₁₇O₁₀⁻: A strongly ferromagnetic stable cluster” has been examined in detail. The authors studied the reactions of iron clusters with oxygen and observed the prominent inertness of this cluster. DFT calculations rationalized its reasonable stability and elucidated the spin accommodation in such a strongly ferromagnetic cluster. However, there are some unresolved and imperfect issues in the manuscript as stated below. Therefore, although this work has been elaborately organized and written, and the results can reasonably support the conclusion, I still hesitate to give the complete agreement for its acceptance of publication in Chem. Commun.

1. The title seems too simple to be attractive for Chem. Commun.

2. The introduction part should be re-organized more logically and more comprehensively.
3. Why authors chose the oxygen content of 10% as the condition of being exposed to O₂. Other contents, like 5% or 20%, should be studied.
4. In Figures 3 and S9, authors should also provide the AIMD simulations of Fe₁₇O₁₀– at 300 K for comparison. At the meantime, the time interval for each step should be added in legends.
5. The potential application(s) of this cluster should be studied.
6. The format of references should be uniform.
7. It is noted that your manuscript needs careful editing with expertise in technical English editing paying particular attention to English grammar, spelling, and sentence structure.

Response to Reviewers' comments

Reviewer: 1

In this manuscript, the authors reported the identification of a set of iron cluster anions from experiment and the corresponding DFT characterization. Experimentally, they observed the distinct activity of the iron clusters against the uptake of molecular oxygen. Although results are very interesting, some points need to be discussed prior to publication.

【Re:】 We thank the referee for the summary and comments on this manuscript. By fully adsorbing all the comments, we have made revisions accordingly.

It is well-known that iron has four naturally-occurring stable isotopes and oxygen has three naturally-occurring stable isotopes, and the corresponding results have never been discussed in the manuscript, especially for the MS spectra, and this is really needed. The combination of these isotopes might explain the noises around the main signal, and more detailed and systematic analysis are needed.

【Re:】 We thank the referee for this helpful comment. Yes, there are isotope distributions of these peaks in the MS spectra, which is helpful to make assignments to the main mass peaks but also a challenge to distinguish the abundant small peaks in the presence of a large quantity of oxygen reactant gas. With a few typical peaks as representative, we have added the comparison of experimental and simulated isotope distributions in both Figure 1 and Supplementary Figure 1. Also, we add such information in the figure caption.

Fig. 1 | Mass spectrometry observation. a, TOF mass spectra of the Fe_n^- clusters produced by the homemade LaVa source. **b** and **c**, Mass spectra of the Fe_n^- clusters after exposure to reactions with different amounts of O_2 (10% in He, 0.1 MPa) corresponding to the on-time

of the pulse valve at 185 μs and 200 μs respectively (namely, the reactant molecule density at 2×10^{19} and 4×10^{19} molecules per cubic meter). The instantaneous pressure of the carrier gas inside the flow tube reactor was approximately 35 Pa, while the lasting time of reaction gas in the reaction tube is estimated to be 1 ms. The vacuum in the source chamber and TOF chamber at $\sim 4.0 \times 10^{-3}$ Pa and $\sim 1.0 \times 10^{-5}$ Pa respectively. The inset shows the simulated and experimental isotope distribution of $\text{Fe}_{17}\text{O}_{10}^-$, $\text{Fe}_{17}\text{O}_{2}^-$, and Fe_{17}^- respectively.

To explain the observed stability, they carried out DFT computations and found very surprising magnetic properties. For the most stable and abundant $\text{Fe}_{17}\text{O}_{10}(-)$ cluster, they calculated the reaction of $\text{Fe}_{17}\text{O}_8(-) + \text{O}_2 = \text{Fe}_{17}\text{O}_{10}(-)$ and a huge reaction energy of about 8 eV, which is as high as that of $\text{Fe}_{17}(-) + \text{O}_2 = \text{Fe}_{17}\text{O}_2(-)$. If $\text{Fe}_{17}\text{O}_{10}(-)$ is really so stable as suggested, the authors must calculate the further reaction of $\text{Fe}_{17}\text{O}_{10}(-) + \text{O}_2 = \text{Fe}_{17}\text{O}_{12}(-)$ to check if this reaction is endergonic or exergonic. If it is endergonic, it is fine with the proposal, however, if it is exergonic, another explanation is needed.

【Re:】 By adsorbing this suggestive comment, we have calculated thermodynamic energy changes from $\text{Fe}_{17}\text{O}_{2x}^-$ to $\text{Fe}_{17}\text{O}_{12}^-$ [$\Delta E = E(\text{Fe}_{17}\text{O}_{2x+2}^-) - E(\text{Fe}_{17}\text{O}_{2x}^-) - E(\text{O}_2)$] and added the information in Figure 4g. Notably, the reaction “ $\text{Fe}_{17}\text{O}_{10}^- + \text{O}_2 \rightarrow \text{Fe}_{17}\text{O}_{12}^-$ ” is also exothermic, but the energy release becomes much smaller (-2.23 eV) than that of “ $\text{Fe}_{17}\text{O}_2^- + \text{O}_2 \rightarrow \text{Fe}_{17}\text{O}_4^-$ ” (-7.03 eV), “ $\text{Fe}_{17}\text{O}_4^- + \text{O}_2 \rightarrow \text{Fe}_{17}\text{O}_6^-$ ” (-7.56 eV), “ $\text{Fe}_{17}\text{O}_6^- + \text{O}_2 \rightarrow \text{Fe}_{17}\text{O}_8^-$ ” (-8.40 eV), and “ $\text{Fe}_{17}\text{O}_8^- + \text{O}_2 \rightarrow \text{Fe}_{17}\text{O}_{10}^-$ ” (-8.45 eV). These calculation results indicate that the formation of $\text{Fe}_{17}\text{O}_{10}^-$ could be significantly faster than its subsequent conversion to $\text{Fe}_{17}\text{O}_{12}^-$.

We have added the description, page 10, first paragraph, line 7-13, also seen as,

“Also, we have calculated the thermodynamic energy changes for the O_2 -addition to the $\text{Fe}_{17}\text{O}_n^-$ ($n = 2, 4, 6, 8$) clusters, as shown in Fig. 4g. Interestingly, the energy gain for $\text{Fe}_{17}\text{O}_{10}^-$ to $\text{Fe}_{17}\text{O}_{12}^-$ (-2.23 eV) is much smaller than that for each step of $\text{Fe}_{17}\text{O}_2^- \rightarrow \text{Fe}_{17}\text{O}_4^- \rightarrow \text{Fe}_{17}\text{O}_6^- \rightarrow \text{Fe}_{17}\text{O}_8^- \rightarrow \text{Fe}_{17}\text{O}_{10}^-$, indicating that the formation of $\text{Fe}_{17}\text{O}_{10}^-$ is significantly faster than its subsequent conversion to $\text{Fe}_{17}\text{O}_{12}^-$.”

As the reaction takes place at given temperature, they should use the thermal corrected Gibbs free energy instead of the computed relative energy to discuss the stability. Probably, temperature plays the decisive role in the stability of the cluster.

【Re:】 We have checked the thermal corrected Gibbs free energies and the zero-point vibration corrected thermodynamic energies, for the typical reaction dynamics of “ $\text{Fe}_{13} + \text{O}_2 \rightarrow \text{Fe}_{13}\text{O}_2$ ”, as shown in Figure S15 of the revised manuscript. It is shown that the calculated Gibbs free energies are slightly higher in values but the energy diagram is almost identical to each other.

Other points

1) The working temperature of the experiment was not given, and according to the DFT results in Figure 4f, the $\text{Fe}_{13}(-)$ reacts with O_2 has a small barrier of 0.54 eV, and this is indeed not very high, and one can expect that at certain temperature, this barrier will vanish forms the oxidized cluster directly. The reaction energy is as large as that of $\text{Fe}_{17}(-)$ about 7-8 eV.

【Re:】 The experiments are performed at room temperature, but the supersonic expansion of helium buffer gas could bring forth cooling effect to the clusters, which is vital to the formation of multinuclear pure metal clusters. Yes, the energetics calculations show that Fe_{13}^- also react with O_2 to form diverse oxide clusters and just need to overcome a small barrier, but the reaction rate is still in sharp contrast to the spontaneous O-O dissociation of $\text{Fe}_{17}\text{O}_2^-$. Notably, the smaller clusters are subject to a relatively smaller vibrational density-of-states (DOS) and shorter lifetime of the vibrational excitation states thus likely simultaneous fragmentation of the unstable/instantaneous $\text{Fe}_n^- \cdot \text{O}_2$ intermediates. We have updated Fig.4 for a more clear comparion between the two systems.

Fig. 4 | Reaction pathway. a/b/c, The typical reaction model of oxygen on the ground-state Fe_{17}^- cluster. d, The typical formation of $\text{Fe}_{17}\text{O}_{10}^-$ through $\text{Fe}_{17}\text{O}_8^-$ reacting with O_2 . e/f, The typical reaction model of oxygen on the ground-state Fe_{13}^- cluster with the end-on and side-on orientation. g, Thermodynamic energy changes from $\text{Fe}_{17}\text{O}_2^-$ to $\text{Fe}_{17}\text{O}_{12}^-$ [$\Delta E = E(\text{Fe}_{17}\text{O}_{2x+2}^-) - E(\text{Fe}_{17}\text{O}_{2x}^-) - E(\text{O}_2)$]. All the optimization was conducted by applying the Gaussian 09 software package, with the BPW91 and the 6-311G(d) basis set for both Fe and O atoms.

2) For DFT, the BPW91 might be B3PW91, please check this carefully. In addition, the authors might check the basis set dependence for the energy and magnetisms by using 6-311+G* basis set (also adding polarization, and this is necessary for both Fe and O atoms, and single-point calculations should be acceptable, at least for the most stable structures).

【Re:】 We thank the referee for this comment. Yes, the BPW91 functional is appropriate in DFT calculations of such systems and we have added a relevant reference (ref.10) in the Supporting Information. For a comparison, we have added calculations of the energy and magnetism of the typical Fe_2O_3^- clusters by using BPW91/6-311G(d), BPW91/6-311+G(d), PBE/6-311G(d), PBE/6-311+G(d), B3LYP/6-311G(d), and B3PW91/6-311G(d), as shown in Supplementary Table 1 (also as attached below), where similar results of both energy and magnetism are indicated.

Supplementary Table 1 | The relative energies and spin multiplicities of Fe_2O_3^- by DFT calculations at BPW91/6-311G(d), BPW91/6-311+G(d), PBE/6-311+G(d), PBE/6-311G(d), B3LYP/6-311G(d), and B3PW91/6-311G(d) level of theory. The spin magnetic moment of Fe_2O_3^- cluster is estimated using the formula $\mu_S = g\sqrt{S(S+1)}\mu_B$ (the Landé factor $g = 2.0023$, a total spin quantum number S).

Fe_2O_3^-		BPW91/6-311G(d)	BPW91/6-311+G(d)	PBE/6-311G(d)	PBE/6-311+G(d)	B3LYP/6-311G(d)	B3PW91/6-311G(d)
Spin multiplicity	Spin magnetic moment (μ_B)	Relative energy (eV)	Relative energy (eV)	Relative energy (eV)	Relative energy (eV)	Relative energy (eV)	Relative energy (eV)
6	6.0	0.04	0.45	0.06	0.40	0.34	0.74
8	8.0	0	0	0	0	0	0
10	10.0	0.83	0.62	0.93	0.60	0.46	0.36

3) At the very beginning of results, they pointed out that they used different doses of oxygen reactants. However, this is not true, since they used always 10% in He.

In summary, major reversion is needed.

【Re:】 We are sorry for this confusion. We used different doses of oxygen reactants (10% in He) by varying on-time of the pulse valve, which is convenient and a common method in cluster reaction experiments.

Reviewer: 2

The paper reports for the first time a very interesting magic iron oxide cluster $\text{Fe}_17\text{O}_{10}$ -among other studied iron oxide anions. Experimental observations are supported by the results of computations using several reliable methods.

I recommend publication after the paper is revised according to the following remarks.

【Re:】 We thank the referee for the positive comments on this manuscript. By fully adsorbing all the comments, we have made major revisions accordingly.

1) Page 2. Line 2 “Typically, FeO is antiferromagnetic” - This is true for the singlet state, but in the ground 5-Delta state, the total spin magnetic moments of both Fe and O atoms coupled parallelly according to the results of DFT-GGA calculations (Theor Chem Acc (2003) 109:298–308).

Line 32. “iron atoms are separated by oxygen atoms but without direct Fe-Fe bonds”. It is not the case for (FeO)_n clusters in cited Ref. [30] where the Fe--Fe bonds present. The cluster (FeO)₁₄ is ferromagnetic, which is difficult to explain.

【Re:】 We thank the referee for pointing this out. By referring to these literatures (including the newly added Ref.14), we have reworded the whole paragraph of the introduction section.

2) Page 3. Line 63. Is it possible that the smaller Fe_n⁻ and Fe_nO_m⁻ are readily coalesce, and this is the reason for the shift into large masses in the mass spectra?

【Re:】 Yes, it is proposed as one of the reasons for some small and unstable Fe_n⁻/Fe_nO_m⁻ clusters to rapidly grow into large ones, such as the formula in the main text,

3) Figure 2. The total spin magnetic moment of the Fe₁₇⁻ is 53 μB. First, the Fe₁₇O₁₀⁻ anion has the odd number of electrons, and this moment should be odd. Second, no increase in the total spin magnetic moment due to oxidation was yet observed to the best of my knowledge. Probably, the increase is because the Fe_n host cluster is in an excited state stabilized by oxygens.

【Re:】 We thank the referee and agree with this comment. Yes, the spin magnetic moment of Fe₁₇⁻ cluster is estimated to be 54 μ_B with a total spin quantum number S=53/2 (i.e., 53 spin-unpaired electrons), but the spin magnetic moment of Fe₁₇O₁₀⁻ cluster is 56 μ_B with a total spin quantum number S=55/2 (i.e., 55 spin-unpaired electrons). The highly ferromagnetic ⁵⁴Fe₁₇⁻ may induce spin-excitation to form the ⁵⁶Fe₁₇O₁₀⁻ cluster. Considering that the ground state of ³O₂ is in spin triplet, this may be easily attained in the oxidation reaction of the Fe₁₇⁻ cluster. We have added this information in the context.

“Notably, the spin magnetic moment of Fe₁₇⁻ cluster is estimated to be 54 μ_B (i.e., 53 spin-unpaired electrons totally), indicative of spin-excitation to form the strongly ferromagnetic Fe₁₇O₁₀⁻ (55 spin-unpaired electrons totally).”

4) Page 7. Line 115. (from -4 to 56 μB), The total spin magnetic moment of a cluster equals to the difference between the number of electrons in the spin majority and spin minority representations, expressed in μB. Thus, no negative value is possible.

【Re:】 We thank the referee for pointing it out. We have revised it accordingly.

5) Figure 4. Only the first step is considered, while the final clusters are $\text{Fe}_{13}\text{O}_8^-$ and $\text{Fe}_{17}\text{O}_{10}^-$. There are many steps ahead, and it is hard to judge on them having only the first step done.

【Re:】 By fully adsorbing the referee's comment, we have managed to optimize the structures of $\text{Fe}_{17}\text{O}_{2m}^-$ ($m=1-6$) and $\text{Fe}_{13}\text{O}_{2m}^-$ ($m=1-5$) clusters, and compared the incremental O_2 -binding energies of $[\Delta E = E(\text{Fe}_n\text{O}_{2m+2}^-) - E(\text{Fe}_n\text{O}_{2m}^-) - E(\text{O}_2)]$, and add these information in Figure 4g and Figure S15, respectively, as well as related discussion shown in page 10, first paragraph, last two sentences, also as seen below,

“In view of this, we have calculated the thermodynamic energy changes for the O_2 -addition to the $\text{Fe}_{17}\text{O}_n^-$ ($n = 2, 4, 6, 8$) clusters, as shown in Fig. 4g. Interestingly, the energy gain for $\text{Fe}_{17}\text{O}_{10}^-$ to $\text{Fe}_{17}\text{O}_{12}^-$ (-2.23 eV) is much smaller than that for each step of $\text{Fe}_{17}\text{O}_2^- \rightarrow \text{Fe}_{17}\text{O}_4^- \rightarrow \text{Fe}_{17}\text{O}_6^- \rightarrow \text{Fe}_{17}\text{O}_8^- \rightarrow \text{Fe}_{17}\text{O}_{10}^-$, indicating that the formation of $\text{Fe}_{17}\text{O}_{10}^-$ is significantly faster than its subsequent conversion to $\text{Fe}_{17}\text{O}_{12}^-$. This could be the reason why $\text{Fe}_{17}\text{O}_{10}^-$ instead of $\text{Fe}_{13}\text{O}_8^-$ is dominated in the reactions of Fe_n^- clusters with oxygen.”

Reviewer: 3

The manuscript entitled “ $\text{Fe}_{17}\text{O}_{10}^-$: A strongly ferromagnetic stable cluster” has been examined in detail. The authors studied the reactions of iron clusters with oxygen and observed the prominent inertness of this cluster. DFT calculations rationalized its reasonable stability and elucidated the spin accommodation in such a strongly ferromagnetic cluster. However, there are some unresolved and imperfect issues in the manuscript as stated below. Therefore, although this work has been elaborately organized and written, and the results can reasonably support the conclusion, I still hesitate to give the complete agreement for its acceptance of publication in Chem. Commun.

【Re:】 We thank the referee for the summary and positive comments on our manuscript. By adsorbing all the referees' comments, we have made major revisions. Hopefully, the revised version will be better to the readers.

1) The title seems too simple to be attractive for Chem. Commun.

【Re:】 We have changed the title to “ *$\text{Fe}_{17}\text{O}_{10}^-$: A strongly ferromagnetic stable cluster with an accordion-like structure*”.

2) The introduction part should be re-organized more logically and more comprehensively.

【Re:】 We thank the referee for this suggestive comment and have re-organized the introduction section accordingly.

3) Why authors chose the oxygen content of 10% as the condition of being exposed to O_2 . Other contents, like 5% or 20%, should be studied.

【Re:】 We thank the referee for this comment. This is simply a compromise consideration of experimental conditions, because too high concentrations make it difficult to distinguish

the reaction process, while too low concentrations may lead to excessive gases which not only affect the cluster distribution profile and even a risk to damage the molecular pump. We have added the results of Fe_n^- clusters reacting with 5% O_2 in He, showing similar results of the size dependence and prominent abundance of $\text{Fe}_{17}\text{O}_{10}^-$, seen as the newly added Fig. S2.

Supplementary Figure 2 | TOF mass spectra of the Fe_n^- clusters produced by the LaVa source (a) and after exposure to the reaction with different amount of O_2 (5% in He), with the on-time of the pulse valve at 195 μs (b), and 220 μs (c), respectively.

4) In Figures 3 and S9, authors should also provide the AIMD simulations of $\text{Fe}_{17}\text{O}_{10}^-$ at 300 K for comparison. At the meantime, the time interval for each step should be added in legends.

【Re:】 We have added the AIMD simulations of $\text{Fe}_{17}\text{O}_{10}^-$ at 300 K for a comparison, and add the time scale of each step (1fs) in the figure caption, as seen in the updated Figure 3 and Figure S10.

Fig. 3 | Thermodynamic phase diagram and AIMD analysis. *a*, Relative ground state formation enthalpies per atom of all the studied $Fe_nO_m^-$ clusters within a convex hull (with respect to Fe_{17}^- and $Fe_2O_3^-$), the points of the set (x, y) are given by $x = \frac{m}{n+m}$; $y = \left[-m \cdot \frac{E_{Fe_2O_3^-}}{3} - \left(n - \frac{2}{3}m \right) \cdot \frac{E_{Fe_{17}^-}}{17} \right] / (n+m)$, corresponding to the total energy of $Fe_nO_m^-$ clusters. The x-axis refers to the atomic number ratio of O relative to the total, while the y-axis shows the relative formation enthalpies per atom above the hull. The cluster structures are given in Supplementary Fig. 6. *b* and *c*, The AIMD simulations of $Fe_{17}O_{10}^-$ at 300 and 800 K for 3250 fs, with the Fe1-O4 distance indicated in Å. The time step was set to 1 fs.

Supplementary Figure 10 | Molecular dynamics simulation. The AIMD simulations of $Fe_{17}O_{10}^-$ at 700 and 900 K for 3500 fs, with the Fe1-O4 distance indicated in Å. The time step was set to 1 fs.

5) The potential application(s) of this cluster should be studied.

【Re:】 We thank the referee for this valuable comment. This strongly ferromagnetic stable cluster $\text{Fe}_{17}\text{O}_{10}^+$ is of potential application as a molecular magnet. We have emphasized this information in the conclusion section.

6) The format of references should be uniform.

【Re:】 We thank the referee for pointing this out. We have fixed the reference format.

7) It is noted that your manuscript needs careful editing with expertise in technical English editing paying particular attention to English grammar, spelling, and sentence structure.

【Re:】 We have conducted a double check of English writing throughout the manuscript.

Again, we thank all the referees for their informative comments which help us to have improved this manuscript significantly.

REVIEWERS' COMMENTS:

Reviewer #1 (Remarks to the Author):

The authors have responded all my questions very satisfactorily, and the manuscript can be recommended for the acceptance for the publication. Congratulation to the authors for their achievement

Reviewer #2 (Remarks to the Author):

The authors correctly responded to my all but one remarks. It seems that our formulation of the spin magnetic moments are somewhat different. The authors consider that the spin magnetic moment is $\mu_S = g\mu_B \sqrt{2S(2S+1)}$ while I was taught that:

“Within the Russell-Saunders scheme, the total magnetic moment μ is defined as $\mu_B(2S + L)$, where L and S are the total angular and spin moments, respectively, and μ_B is the Bohr magneton. If one neglects contributions from L , then the total magnetic moment becomes $\mu = g\mu_B S$, where the gyromagnetic ratio g is 2.0023. This definition of μ corresponds to the Heisenberg approximation. Rounding the gyromagnetic ratio, the scalar total magnetic moment $\mu_S = 2S\mu_B = [n_\alpha - n_\beta]\mu_B$, where n_α and n_β are the numbers of the electrons in the α - and β -spin representations, respectively”.

The authors should check if their formulation is correct.

Reviewer #3 (Remarks to the Author):

The authors have answered the questions raised by the reviewers and revised their manuscript accordingly. Hence, the manuscript can be accepted in this form.

Response to Reviewers' comments

Reviewer: 1

The authors have responded all my questions very satisfactorily, and the manuscript can be recommended for the acceptance for the publication. Congratulation to the authors for their achievement.

【Re:】 We thank the reviewer for the positive comments.

Reviewer: 2

The authors correctly responded to my all but one remarks. It seems that our formulation of the spin magnetic moments are somewhat different. The authors consider that the spin magnetic moment is $\mu_S = g\mu_B \sqrt{[2S(2S+1)]}$ while I was taught that:

“Within the Russell-Saunders scheme, the total magnetic moment μ is defined as $\mu_B(2S + L)$, where L and S are the total angular and spin moments, respectively, and μ_B is the Bohr magneton. If one neglects contributions from L , then the total magnetic moment becomes $\mu = g\mu_B S$, where the gyromagnetic ratio g is 2.0023. This definition of μ corresponds to the Heisenberg approximation. Rounding the gyromagnetic ratio, the scalar total magnetic moment $\mu_S = 2S\mu_B = [n\alpha - n\beta]\mu_B$, where $n\alpha$ and $n\beta$ are the numbers of the electrons in the α - and β -spin representations, respectively”.

The authors should check if their formulation is correct.

【Re:】 We thank the referee for the positive comments. Regarding the concept of spin magnetic moment, the referee is right. Yes, there is a slight difference in introducing this concept in the physics textbook and chemistry books. Accordingly we have added a reference of the commonly used chemistry textbooks (Ref.11 in ESI).

Reviewer: 3

The authors have answered the questions raised by the reviewers and revised their manuscript accordingly. Hence, the manuscript can be accepted in this form.

【Re:】 We thank the reviewer for the positive comments.

Again, we thank all the referees for their positive comments and instruction which help us to have improved this manuscript significantly.